# The Impact of a Large-Scale Social and Behavior Change Communication Intervention in the Lake Zone Region of Tanzania on Knowledge, Attitudes, and Practices Related to Stunting Prevention

**DOI:** 10.3390/ijerph20021214

**Published:** 2023-01-10

**Authors:** Kirk Dearden, Generose Mulokozi, Mary Linehan, Dennis Cherian, Scott Torres, Joshua West, Benjamin Crookston, Cougar Hall

**Affiliations:** 1Corus International/IMA World Health, 1730 M St NW #1100, Washington, DC 20036, USA; 2Corus International/IMA World Health, Nyalali Curve, Plot 1657, Dar es Salaam P.O. Box 9260, Tanzania; 3RTI International, 701 13th St NW #750, Washington, DC 20005, USA; 4Department of Public Health, Brigham Young University, LSB, Provo, UT 84602, USA

**Keywords:** SBCC, Tanzania, stunting, public health

## Abstract

Background: Large-scale social and behavioral change communication (SBCC) approaches can be beneficial to achieve improvements in knowledge, attitudes, and practices (KAP). Addressing Stunting in Tanzania Early (ASTUTE) included a significant SBCC component and targeted precursors to stunting including KAP related to maternal and child health, antenatal care, WASH, childhood development, and male involvement. METHODS: Baseline, midline, and endline surveys were conducted for a total of 14,996 female caregivers and 6726 male heads of household in the Lake Zone region of Tanzania. Regression analyses were used to estimate differences in KAP from baseline to midline and endline. Results: Women’s knowledge of handwashing and infant/child feeding practices, and attitudes related to male involvement, consistently improved from baseline to midline and baseline to endline. Women’s practices related to antenatal care, breastfeeding, and early child development improved from baseline to midline and baseline to endline. Improvements in KAP among male heads of household were varied across indicators with consistent improvement in practices related to child feeding practices from baseline to midline and baseline to endline. Conclusion: Many changes in KAP were observed from baseline to midline and baseline to endline and corresponded with SBCC programming in the region. These results provide support for the value of large SBCC interventions. Public health efforts in settings such as Tanzania may benefit from adopting these approaches.

## 1. Introduction

Tanzania has achieved dramatic improvements in maternal and child health in recent decades, yet undernutrition remains a serious public health problem [1]. Despite the Tanzanian government providing free, universal maternal, newborn, and child health (MNCH) services, women’s dietary practices and nutritional status are far from ideal. In Tanzania, challenges include an uneven commitment to women’s nutrition, limited human resources, and a lack of exposure to innovative social and behavioral change communication (SBCC) strategies to improve nutrition practices [2]. SBCC interventions are designed to address social and behavioral issues generally [3] and have been widely implemented for childhood nutrition and stunting prevention purposes specifically [4,5,6,7]. SBCC interventions may be inclusive of multiple methods including mass communication, interpersonal communication, group-based approaches, advocacy, community or social mobilization, and capacity strengthening [8]. Recent systematic reviews and meta-analyses of SBCC approaches to improve child nutritional status mostly demonstrate the effectiveness of such programs, particularly when a clear behavior change objective has been targeted [4,6,7]. Addressing Stunting in Tanzania Early (ASTUTE) was a large-scale, comprehensive, five-year SBCC program inclusive of mass media communications (e.g., television, radio); interpersonal communication (IPC) interventions (e.g., home visits, home-based health education); group-based approaches (e.g., support groups for women, positive deviance/hearth, community mobilization days; community mobilization, community outreach; health facility-based counseling; and multisectoral capacity building [e.g., regional and district government staff, health facility workers, community health workers]). ASTUTE activities focused on standard MNCH indicators, including making regular health facility visits to receive antenatal care (ANC); infant and young child feeding (IYCF) practices; water, sanitation, and hygiene (WASH) practices; measures of early childhood development (ECD); and indicators of male involvement.

IMA World Health, along with consortium partners, implemented ASTUTE in Tanzania’s Lake Zone region. The program included training more than 6000 community health workers, facility workers, staff, and volunteers from 50 civil society organizations to provide leadership and interventions in creating cultural and behavioral shifts in key MNCH-related practices. Mass media efforts included radio and television spots disseminating six key message themes: (1) maternal health and nutrition during pregnancy; (2) exclusive breastfeeding for children 0–6 months; (3) complementary feeding for children 6–24 months; (4) early child development; (5) water, sanitation, and hygiene practices; and (6) diarrhea treatment. IPC programming included home visits from trained community health workers (CHWs) and organized mobile outreach. Group-based approaches included positive deviance/hearth groups and support groups. IMA partnered with Development Media International (DMI)—its partner in the development and implementation of radio and TV spots—to design and conduct baseline, midline, and endline surveys to inform program direction and to assess impact during, in the middle, and at the end of the program. Numerous studies exploring the associations of key health outcomes and exposure to core program elements and indicators at either baseline, midline, or endline have been published previously [9,10,11,12,13,14,15,16,17,18]. Associations between knowledge of key health-promoting behaviors included in SBCC programming and stunting prevention have been significant [10,13,15]. Moffat et al. [14] examined individual characteristics associated with having experienced exposure to SBCC programming. Findings indicated that a woman’s increased wealth, ownership of a cell phone, access to radio and TV, increased opportunity for household decision-making, and support from a husband were predictive of SBCC mass media exposure, but not IPC components of the SBCC. Broadbent et al. [11] identified SBCC as an effective approach for the promotion of ECD knowledge and behaviors. In particular, a mother’s exposure to the SBCC’s IPC programming was positively associated with all measured ECD behaviors, including talking to, drawing with, counting with, naming objects with, and playing with your child. Similarly, previous analyses of ASTUTE data have also explored the impact of key SBCC programming related to male involvement, including maternal perceptions of the role of men during pregnancy [16] and associations between men’s engagement in household chores and both maternal health and ANC-seeking behaviors [12]. However, to date, no study has provided a comprehensive analysis inclusive of both mass media and IPC programming using baseline, midline, and endline ASTUTE data. The purpose of this study was to examine if this large-scale SBCC intervention was associated with changes in KAP related to key MNCH indicators comparing midline and endline data to baseline data.

## 2. Materials and Methods

UKaid and the Foreign, Commonwealth and Development Office (FCDO) provided funding to IMA World Health for the implementation of ASTUTE. A consistent tagline was used at the end of each theory-based radio spot, which was broadcast a total of 70,000 times. TV spots were aired before and during the evening news on national and regional stations a total of 1198 times. CHWs used a problem-based negotiated behavior change approach during in-home visits to implement IPC components of the intervention. They counseled mothers and referred children with growth faltering to health facilities for treatment and counseling. They also encouraged both mothers and male partners to engage in stimulation activities (e.g., drawing, playing, playing, naming objects, or talking with them) for their children by providing education and support.

### 2.1. Sample

Data came from three distinct cross-sectional surveys completed by a unique sub-sample of participants during each data collection period between 2016 and 2020. Surveys were conducted in five regions of Tanzania’s Lake Zone region, namely Geita, Kagera, Kigoma, Mwanza, and Shinyanga. A stratified, multi-stage random sample design was used to select survey participants. Eligibility to participate was limited to households with a child under two years of age. Participants were randomly sampled from 243 villages that were selected from among the five participating regions. The baseline survey was carried out in 2016, prior to the launch of ASTUTE programming. A total of 5000 mothers, hereafter known as female caregivers, and 1144 corresponding fathers, hereafter known as male heads of household, were surveyed. The midline survey was conducted in 2019 and included 5000 female caregivers and 2502 male heads of households. The endline survey was conducted in 2020 after all ASTUTE programming ended and included 4996 female caregivers and 3080 male heads of household. The present study sample includes all baseline, midline, and endline participants for a total of 14,996 female caregivers and 6726 male heads of household. Participant demographics are presented in Table 1.

### 2.2. Study Design and Procedure

The female caregiver of the youngest child in the home responded to questionnaire items. The male head of household was asked to respond only if available and applicable. IPSOS, a local research firm, collected all three waves of data. They comprised a field team with 50 enumerators and 10 supervisors. Twenty-five percent of records were quality-checked using revisits and phone checks. DMI’s internal IRB and Tanzania’s National Institute for Medical Research (TZ: NIMR/HQ/R.8a/Vol.IX/2344) provided Institutional Review Board (IRB) approval. Informed consent was collected before the surveys began and participants were reminded that participation was voluntary and they could stop the survey at any time. Questionnaire items were written in English, translated into Kiswahili, and then back-translated to English to ensure the original meaning was retained. The questionnaires were piloted, modified, and finalized before being administered to participants. Interviews were conducted in the participants’ homes and lasted on average 50–60 min. Baseline data were collected using hard copies and midline and endline data were recorded using smartphones and PDAs (personal digital assistants).

### 2.3. Measures

Participants’ demographic characteristics were measured and collected. Exposure to the various components of the intervention (radio, TV, and IPC intervention in the midline and endline questionnaires) were also collected along with key MNCH indicators.

Wealth. A calculated composite variable adapted from a previously validated index was used to estimate household wealth [19]. Two sub-indices comprised the index. The first sub-index represented access to services and ownership of consumer durables was the second. Items pertaining to access to services included the availability of safe drinking water sources (e.g., protected wells, a public standpipe) and safe sanitation (e.g., a flush toilet). Pit latrines were not considered safe sanitation for this study. Seven items were measured to represent consumer durables. These included ownership of a radio, TV, bicycle, motorcycle, mobile phone, boat, or animal-drawn cart. Each index was calculated by summing the total of the indicators within each index. An average of the two indices was then used to calculate an overall wealth score, with possible values ranging between 0 and 1. Higher wealth scores indicate higher socioeconomic status. Housing quality was not included in this index as the data were not available.

Intervention Exposure. Only data collected at endline were used to measure exposure to the intervention. Exposure was estimated separately for each of the radio, TV, and IPC intervention components. Exposure to the radio component was coded ‘yes’ if respondents reported affirmatively to having heard the example spot(s) that concluded with the sound of a laughing baby or if they reported having heard radio messages that advised about maternal and child health and/or child development. Exposure to TV was coded ‘yes’ if respondents reported affirmatively to having seen the example image frame(s) on TV or ‘reported seeing messages on the TV that advised about maternal/child health/child development’. IPC exposure was coded ‘yes’ if respondents reported affirmatively that a (community) health worker had visited their home and advised them about maternal and child health and/or child development. Exposure to each intervention component (radio, TV, and IPC) was measured for female caregivers. IPC mostly targeted female caregivers, so male head of household respondents were only asked questions about exposure to radio and TV.

### 2.4. Analysis

Data were deidentified and shared only with study personnel to ensure confidentiality. STATA version 16 (College Station, TX, USA) was used to clean and recode variables in each of the three datasets. SAS 9.4 (Cary, NC, USA) was used to conduct analyses. Basic frequency statistics were calculated for key demographic variables. Logistic regression analysis was used to identify changes in KAP at each time point, comparing the midline and endline values to the baseline values. All models were adjusted for respondent age, education, and household wealth.

## 3. Results

### 3.1. Demographics

Most female caregivers at each round were 20–29 years of age, able to read, had completed primary school, were crop farmers, and were monogamous (Table 1). Kiswahili and Sukuma were the most spoken languages. Almost half (46%) of female caregivers reported hearing mass media messages only regarding the intervention, while 14 percent heard both mass media messages and received IPC, 6 percent received IPC only, and 34 percent had no reported exposure to the intervention (Table 2).

**Table 1 ijerph-20-01214-t001:** Female Caregiver Characteristics by Survey Round.

		Baseline		Midline		Endline	
		*n*	%	*n*	%	*n*	%
Mothers							
Age (years)	<20	22	(0.5)	92	(1.8)	93	(1.9)
	20–29	2409	(55.3)	2697	(54.0)	2865	(57.4)
	30–39	1183	(27.2)	1357	(27.2)	1352	(27.1)
	40+	290	(6.7)	305	(6.1)	273	(5.5)
	Missing	647	(12.9)	64	(1.3)	5	(0.1)
Primary language	Swahili	1848	(37.0)	2005	(40.2)	1672	(33.5)
	Sukuma	1357	(27.1)	1481	(29.7)	1652	(33.1)
	Other	1795	(35.9)	1506	(30.2)	1669	(33.4)
Able to read ^1^	No	1316	(26.3)	1208	(24.2)	1092	(21.9)
	Yes	3684	(73.7)	3784	(75.8)	3901	(78.1)
Occupation	Crop farmer	3576	(71.5)	3429	(68.7)	3495	(70.0)
	Other	1424	(28.5)	1563	(31.3)	1498	(30.0)
Completed primary school	No	1546	(30.9)	1489	(29.8)	1480	(29.6)
	Yes	3454	(69.1)	3503	(70.2)	3513	(70.4)
Marital Status ^2^	Single	293	(5.9)	273	(5.5)	211	(4.2)
	Monogamous	3549	(71.0)	3474	(69.6)	3848	(77.1)
	Polygamous	291	(5.8)	506	(10.1)	490	(9.8)
	Other	867	(17.3)	739	(14.8)	444	(8.9)
Total		5000	(100)	4992	(100)	4993	(100)
Fathers							
Age (years)	<20	7	(0.01)	88 (3.5)		5	(0.4)
	20–29	335	(33.2)	812 (32.5)		360	(31.7)
	30–39	392	(38.9)	1025 (40.9)		452	(39.8)
	40+	275	(27.3)	577 (23.1)		318	(28.0)
Occupation	Informal	991	(98.2)	2377	(95.1)	1099	(96.1)
	Formal	18	(1.8)	125 (4.9)		45	(3.9)
Completed primary school		205	(20.1)	550 (21.9)		215	(18.8)
		806	(79.9)	1952 (78.1)		929	(81.2)
Total		1009		2502		1135	

^1^ Defined as being able to read aloud some or all of the sentence, “Unaweza kusoma na kuandika”. ^2^ Other includes informal union, widowed, divorced, or separated.

### 3.2. Knowledge and Attitudes

Women’s knowledge about when to begin giving complementary foods (foods and liquids in addition to breastmilk) and critical handwashing moments improved significantly from baseline to midline and baseline to endline (Table 3). While knowledge levels increased slightly from baseline to midline and baseline to endline for early initiation of breastfeeding and exclusive breastfeeding for the first six months of life, increases were not statistically significant. Understanding that handwashing without soap does not clean hands properly increased significantly from baseline to midline but the increase from baseline to endline was not significant.

### 3.3. Practices

Female caregivers reported eating significantly more types of food, attending more antenatal visits, having a partner help with chores during the most recent pregnancy, singing more to the child, and drawing more with the child from baseline to midline and baseline to endline (Table 4). Female caregivers were significantly more likely to report emptying both breasts when breastfeeding, counting in front of the child more, and engaging in more activities with the child from baseline to endline, but not from baseline to midline. Inversely, female caregivers increased breastfeeding in the first hour of life from baseline to midline but not from baseline to endline.

Male heads of household were significantly more likely to feed the youngest child in the previous three months from baseline to midline than from baseline to endline (Table 5). Male heads of household were significantly more likely to point out objects to the child and talk to and play with the child from baseline to endline but not from baseline to midline. There were significantly more male heads of household who reported helping their wives with chores during a previous pregnancy from baseline to midline but not from baseline to endline.

## 4. Discussion

The purpose of this study was to determine whether MNCH indicators of KAP improved from baseline to midline and baseline to endline. While not all changes were significant, and not all changes were consistent at the different data collection periods, the results were positive overall. Both midline and endline results for 26 indicators were compared to baseline, for a total of 52 comparisons. Of these 52 comparisons, 33 (63%) demonstrated statistically significant improvement. For female caregivers, improvement in both knowledge and attitudes was less impressive than improvements among practices indicators. For example, of the eight knowledge and attitudes indicators, four (50%) showed significant improvement at the midline compared to the baseline, and four (50%) significantly improved at the endline compared to the baseline. It should be noted that three indicators showed significant improvement at both the midline and endline compared to baseline, while one indicator was significant at the midline only and one significant at the endline only when compared to baseline. More importantly, of the 12 practices indicators measured for female caregivers, nine (75%) significantly improved by midline, and 10 (83%) significantly improved by endline when compared to baseline. As with the knowledge and attitude indicators, the majority of practice indicators improved in comparison to baseline at both midline and endline, yet one indicator showed significant improvement only at the midline while two showed significant improvement only at the endline when compared to baseline. Knowledge about exclusive breastfeeding and the introduction of foods and liquids, while similar, appear to be interpreted differently among respondents. Evidence of this is that nearly 92 percent of respondents at baseline were already knowledgeable about exclusive breastfeeding, while only 72 percent were knowledgeable about when to introduce foods and liquids into a child’s diet. Among the five practices measured for male heads of household, three (60%) had significantly improved by midline and three (60%) had significantly improved by endline when compared to baseline. As with measures for female caregivers, significant improvements among male heads of household were not consistent across data collection periods. Other studies of large-scale health-related campaigns in low- and middle-income countries have reported similar encouraging results [20,21,22]. The purpose of these campaigns is to share information about optimal health behaviors, improve attitudes, and get large numbers of individuals to adopt these behaviors [23]. Well-implemented media campaigns have been shown to change social norms and behaviors in positive ways [23,24,25,26].

The current study’s findings are especially impressive given the difficulty in promoting behavior change and improving health practices. While there were fewer improvements in knowledge and attitudes, it is important to note that efforts to increase knowledge and attitudes are undertaken solely with the express desire to improve practices.

Ideally, there would be consistent and linear improvement in the practice of all targeted behaviors over time when comparing midline and endline data to baseline data. Including the same participants at each data collection period, as opposed to surveying a unique sample of participants at each period as this study did, may have yielded more consistent results. Variations in the improvement of different indicators at different time periods in this study are consistent, however, with other SBCC programming [6]. In their review of SBCC programming’s impact on specific nutrition-related indicators, Kennedy et al. note that varying levels of success among indicators are common when evaluating large-scale SBCC interventions [6]. Evaluation of SBCC interventions is impeded by the challenge of measuring and determining the dose and response to various program interventions. Kennedy et al. indicate that combining individuals with various levels of exposure to interventions is a common practice hindering analyses of SBCC [6]. Indeed, the current study did not attempt to correlate SBCC exposure to study indicators nor did it quantify an individual participant’s exposure (dose) to mass media or IPC programming with respect to KAP surrounding MNCH (response), rather it addressed the combined impact of SBCC programming on the population represented by a large, randomly selected cohort at midline and endline compared with a similar cohort at baseline. For these reasons, the current study’s level of variation in results among variables and between time periods may be expected.

Several significant improvements in knowledge and attitudes were related to male involvement. For example, female caregivers reported that male involvement with household chores during pregnancy increased at both midline and endline compared to baseline. Female caregivers also reported that men would approve of other men helping in this way. Similarly, male heads of household perceived that most men in the community helped wives with household chores during pregnancy at midline compared to baseline. These shifts in knowledge and attitudes are especially promising if they are indicative of shifting societal and gender norms leading to greater equity in gender relations through male involvement in household and parenting duties [27].

It is noteworthy that those knowledge and attitudes indicators which did not show significant improvements from baseline to midline or baseline to endline were generally already very high at baseline (i.e., *Agree child should only be given breastmilk for first 6 months;* and *How soon after birth should a child be put to the breast?*) rendering any additional significant improvements in the population statistically challenging. For example, in their analysis of endline-only ASTUTE data, Beckstead et al. [9] found significant associations between exposure to SBCC radio programming and a variety of IYCF practices as well as significant associations between SBCC television programming and IYCF knowledge. Significant findings between SBCC exposure and enline data from Beckstead et al. may be helpful in gauging the impact of SBCC programming on indicators that are high at each of the three data collection periods in the current study.

Improvements in practices were impressive, especially among female caregivers. Only three practices indicators did not show significant improvement from baseline to midline (i.e., *Usually empty both breasts when breastfeeding; Count in front of child last week*; and *Mean number of activities engaged with child in last week*), and only two practices indicators did not show significant improvement from baseline to endline (i.e., *Initiated breastfeeding in first hour* and *Mean number of activities engaged with child in last week*).

There were significant improvements in ECD-related KAP in the current study. The current study’s findings might be compared with the cross-sectional analysis of endline-only data by Broadbent et al. [11] who identified a significant association between exposure to SBCC mass media programming and ECD behaviors. Early childhood development and cognitive stimulation are vital to the child’s well-being and have been found to impact both physiological [28] and neural development [29]. For example, an investigation of parental involvement and ECD in Tanzania concluded that higher levels of parental stimulation resulted in improved child cognition, language, and motor skills [30]. As a stunting prevention approach, ECD has been found to be equally important as traditional nutritional and dietary indicators [31,32]. SBCC appears to be an effective approach for increasing female caregiver ECD as measured by singing to a child and drawing with a child in particular.

Improvements in practices indicators of male heads of household were modest, with “*Man helped feed child frequently in past three months*” showing significant improvement compared to baseline at both midline and endline. Other indicators only saw significant improvements between baseline and midline (i.e., *Male helped wife with chores during pregnancy*) or baseline to endline (i.e., *Man points out objects to child;* and *Male caregiver talked to the child and played with the child in the last week*). “*Man purchased food for child in past month*” did not significantly improve at either midline or endline when compared to baseline. While only modest differences in positive directions were observed, these findings can be interpreted as progress given the challenge of increasing male involvement generally and increasing male participation in promoting early childhood development specifically. Predominant and prevailing sociocultural norms work to discourage male involvement in pregnancy and child-rearing practices [33]. Extensive literature has documented the challenge of overcoming lingering cultural beliefs that childcare is the role of women only and that the role of men is to provide financially for the family [34,35,36,37]. Well-intended public health interventions have perhaps indirectly and unintentionally reinforced such norms through programming focused exclusively on mothers at the exclusion of fathers [35,38,39,40].

It is worth reiterating that the current study found comparatively fewer improvements in knowledge and attitude indicators as compared to practices indicators. Altering behavior is generally more challenging than increasing knowledge or shifting existing attitudes toward increasing care or concern for specific health behaviors or practices. Indeed, SBCC programs often measure shifts in knowledge and attitudes as a proxy for the more difficult to influence and measure practices. Future research should continue to explore the complex and weaker-than-expected association between an individual’s knowledge and his or her behavioral practices [41,42].

### Limitations

Evaluating large-scale SBCCs can be challenging for a variety of reasons and the current study’s findings should be considered in light of several limitations. Levels of exposure to SBCC programming were reported (Table 2) to show the reach of this programming, but the current study did not attempt to correlate SBCC exposure to study indicators. When a region is flooded with health promotion messaging, calculating both direct or indirect exposure and measuring the dose or duration of said exposure to that messaging is difficult and often beyond the scope of program planners, implementers, and researchers. For example, household discussions between one family member exposed to SBCC programming and another without exposure may extend the program’s reach. Such indirect exposure is key to the success of SBCC approaches while paradoxically increasing the challenge of program evaluation. Additionally, the measurement of several study indicators was impacted by unusually high scores at baseline. Finally, given that the ASTUTE program was implemented in only five regions, other regions are not represented. Despite these limitations, this study suggests a pattern of improvements in KAP after ASTUTE began. These conclusions are based on rigorous methods, a large sample size, and strict data collection regimens across three sampling periods. Findings are supportive of future SBCC interventions, especially programming targeting knowledge and attitudes related to male involvement and IYCF together with practices related to ANC, ECD, and male involvement. Additional research is needed to better understand how large-scale communication campaigns can be improved and integrated into other health promotion efforts and interventions. Further investigation is likewise needed in understanding why some indicators examined in this study remain resistant to SBCC messaging. Identifying SBCC approaches capable of effectively promoting ECD among male heads of household is of prime importance. Promoting the benefits of breastfeeding practices and a woman’s reduced workload during pregnancy are examples of two other important practices in need of continued attention.

## 5. Conclusions

This study examined KAP related to standard MNCH indicators before, during, and after a large-scale SBCC program designed to address the persistent challenge of childhood stunting in Tanzania. Data analyzed compared a large sample of participants at baseline with similarly sized samples at both midline and endline. Study results support the use of SBCC programming for improving KAP generally. SBCC programming appears particularly effective at influencing the knowledge and attitudes of female caregivers related to male involvement and IYCF. SBCC approaches are similarly effective in promoting practices associated with ANC, ECD, and male involvement. Female caregiver knowledge of breastfeeding timing together with attitudes related to women doing chores appear to be resistant to SBCC programming. These findings can help to inform future SBCC interventions targeting key indicators associated with stunting prevention.

## Figures and Tables

**Table 2 ijerph-20-01214-t002:** Program Exposure at Endline.

	*n*	%
Female Caregivers		
None	1704	34.1
IPC Only ^1^	284	5.7
Media Only ^2^	2312	46.3
IPC and Media ^3^	696	13.9
Male Heads of Household		
None	900	29.4
Radio or TV ^2^	2164	70.6

^1^ Responded ‘yes’ to having received an in-home visit from a CHW who advised about maternal/child health/child development. ^2^ Responded ‘yes’ to hearing radio spot(s) ending with a laughing baby sound, hearing messages that advised about maternal/child health/child development, or ‘reported seeing messages on the TV that advised about maternal/child health/child development’. ^3^ Exposure to radio, TV, and IPC was only estimated for female caregivers.

**Table 3 ijerph-20-01214-t003:** Logistic Regression Models for Female Caregiver Knowledge and Attitudes.

Outcome	Round	Total	*n*	%	Adj. OR	95% CI	*p* Value (LRT)
Answer ‘no’ to “Does hand washing with water alone make your hands clean?”	Baseline	5000	4346	86.9	Ref	Ref	Ref
	Midline	5000	4651	93.0	2.20	(1.631, 2.971)	0.000 *
	Endline	4996	4453	89.1	1.18	(0.749, 1.848)	0.480
Believes that women should do fewer chores during pregnancy	Baseline	4655	1593	34.2	Ref	Ref	Ref
	Midline	5000	2268	45.4	1.54	(0.910, 2.612)	0.108
	Endline	4996	1925	38.5	1.21	(0.703, 2.100)	0.486
Believes at least some female friends receive help from male partners with household chores during pregnancy ^1^	Baseline	5000	3454	69.1	Ref	Ref	Ref
	Midline	5000	4382	87.6	3.89	(2.790, 5.424)	0.000 *
	Endline	4996	4668	93.4	6.16	(4.423, 8.581)	0.000 *
Feels that men will approve of other men in the community who help wives with household chores during pregnancy ^1^	Baseline	5000	2340	46.8	Ref	Ref	Ref
	Midline	5000	3634	72.7	3.05	(2.053, 4.520)	0.000 *
	Endline	4996	3451	69.1	2.53	(1.880, 3.387)	0.000 *
Agree child should only be given breastmilk for first 6 months	Baseline	5000	4592	91.8	Ref	Ref	Ref
	Midline	4996	4619	92.5	1.08	(0.753, 1.535)	0.689
	Endline	4996	4606	92.2	1.23	(0.913, 1.660)	0.174
Report 6 months when asked when child should be given other foods/liquids	Baseline	5000	3633	72.7	Ref	Ref	Ref
	Midline	5000	4226	84.5	2.13	(1.397, 3.255)	0.000 *
	Endline	4996	4349	87.1	2.52	(1.641, 3.882)	0.000 *
How soon after birth should a child be put to the breast ^3^	Baseline	5000	4149	83.0	Ref	Ref	Ref
	Midline	5000	4253	85.1	1.12	(0.743, 1.626)	0.636
	Endline	4996	4417	88.4	1.53	(0.833, 2.824)	0.170
Outcome	Round	Total	M	S.D.	Beta	95% CI	*p* Value (LRT)
Mean number of times female caregiver identified as critical for handwashing ^2^	Baseline	5000	2.92	1.5	Ref	Ref	Ref
	Midline	5000	3.34	1.4	0.49	(−0.186, 1.164)	0.114
	Endline	4996	3.55	1.4	0.59	(0.301, 0.886)	0.005 *

Model controlled for participant age, education, and wealth. OR = Odds Ration, CI = Confidence Interval, LRT = Likelihood Ratio Test, Ref = Reference. * = Statistically significant. ^1^ Chores including fetching water, farming, or ‘something else so that you could rest’. ^2^ Times included: After latrine use; after assisting a child who has defecated; before preparing food; before eating; before feeding a child; after cleaning the compound; after contact with animal feces. ^3^ Within the first hour.

**Table 4 ijerph-20-01214-t004:** Logistic Regression Models for Female Caregiver Practices.

Outcome	Round	Total	N	%	Adj. OR	95% CI	*p* Value (LRT)
Mother ate more types of food during last pregnancy	Baseline	4648	331	7.10	Ref	Ref	Ref
	Midline	4930	1200	24.3	4.43	(3.379, 5.794)	0.000 *
	Endline	4959	979	19.8	3.14	(2.266, 4.319)	0.000 *
Obtained tablets or syrup during last pregnancy	Baseline	4648	3535	76.1	Ref	Ref	Ref
	Midline	4932	4222	85.6	2.13	(1.244, 3.645)	0.006 *
	Endline	4957	4388	88.5	2.41	(1.394, 4.178)	0.002 *
Attended antenatal care during last pregnancy	Baseline	4642	1584	34.1	Ref	Ref	Ref
	Midline	4934	3905	79.1	7.91	(4.773, 13.106)	0.000 *
	Endline	4957	4088	82.5	9.10	(4.149, 19.970)	0.000 *
Attended antenatal care 4+ times during last pregnancy	Baseline	5000	748	15.0	Ref	Ref	Ref
	Midline	5000	1461	29.2	2.45	(1.778, 3.366)	0.000 *
	Endline	4996	1094	21.9	1.59	(1.011, 2.499)	0.045 *
Usually empty both breasts when breastfeeding (if currently breastfeeding)	Baseline	4014	3597	89.6	Ref	Ref	Ref
	Midline	4337	3920	90.4	1.12	(0.805, 1.566)	0.680
	Endline	4392	4138	94.2	1.96	(1.013, 3.779)	0.046 *
Worked less during the last pregnancy	Baseline	5000	2776	55.5	Ref	Ref	Ref
	Midline	5000	3441	68.8	2.05	(1.466, 2.864)	0.000 *
	Endline	4996	3351	67.1	1.61	(1.138, 2.285)	0.007 *
Initiated breastfeeding in first hour	Baseline	5000	3439	69.9	Ref	Ref	Ref
	Midline	5000	3765	75.3	1.47	(1.122, 1.928)	0.005 *
	Endline	4996	3331	66.7	0.88	(0.525, 1.434)	0.580
Wife reported that husband frequently helped with chores during last pregnancy	Baseline	5000	1897	37.9	Ref	Ref	Ref
	Midline	5000	2473	49.5	1.84	(1.513, 2.242)	0.000 *
	Endline	4996	2487	49.8	1.60	(1.325, 1.920)	0.000 *
Sung to child in last week	Baseline	5000	3598	72.0	Ref	Ref	Ref
	Midline	5000	4234	84.7	2.09	(1.719, 2.543)	0.000 *
	Endline	4996	4475	89.6	3.33	(2.324, 4.780)	0.000 *
Draw with child in last week	Baseline	5000	658	13.2	Ref	Ref	Ref
	Midline	4998	935	18.5	1.42	(1.047, 1.927)	0.024 *
	Endline	4993	1225	24.5	2.12	(1.453, 3.091)	0.000 *
Count in front of child in last week	Baseline	5000	1317	26.3	Ref	Ref	Ref
	Midline	4996	1479	29.6	1.20	(0.972, 1.459)	0.092
	Endline	4996	1926	38.6	1.71	(1.404, 2.077)	0.000 *
Outcome	Round	Total	M	S.D.	Beta	95% CI	*p* Value (LRT)
Mean number of activities female caregiver engaged with child in last week	Baseline	4959	4.8	1.91	Ref	Ref	Ref
	Midline	4985	5.3	1.79	0.43	(−0.042, 0.896)	0.065
	Endline	4993	5.73	1.71	0.91	(0.353, 1.448)	0.011

Model controlled for participant age, education, and wealth. OR = Odds Ration, CI = Confidence Interval, LRT = Likelihood Ratio Test, Ref = Reference. * = Statistically significant.

**Table 5 ijerph-20-01214-t005:** Logistic Regression Models for Male Heads of Household Knowledge, Attitudes and Practices.

Outcome	Round	Total	N	%	Adj. OR	95% CI	*p* Value (LRT)
Man points out objects to child	Baseline	1143	701	61.3	Ref	Ref	Ref
	Midline	2502	1669	66.7	1.31	(0.979, 1.747)	0.069
	Endline	3080	2335	75.8	1.91	(1.700, 2.149)	0.000 *
Man purchased food for child in past month	Baseline	1143	634	55.5	Ref	Ref	Ref
	Midline	2502	1267	50.6	0.84	(0.616, 1.142)	0.282
	Endline	3078	1711	55.6	0.98	(0.778, 1.230)	0.844
Man helped feed child frequently in past three months	Baseline	1144	308	26.9	Ref	Ref	Ref
	Midline	2448	799	32.6	1.34	(1.066, 1.693)	0.012 *
	Endline	3048	1121	36.8	1.55	(1.025, 2.345)	0.038 *
Male caregiver talked to the child and played with the child in the last week.	Baseline	1142	587	51.4	Ref	Ref	Ref
	Midline	2502	1448	57.9	1.3	(0.902, 1.865)	0.161
	Endline	3079	2110	68.5	2	(1.692, 2.370)	0.000 *
Male helped wife with chores during pregnancy	Baseline	1144	662	57.9	Ref	Ref	Ref
	Midline	2448	1672	66.8	1.36	(1.005, 1.846)	0.046 *
	Endline	3048	1834	59.7	1.09	(0.783, 1.512)	0.616
Perceive that half or more men in community help wives with chores during pregnancy	Baseline	1144	157	13.7	Ref	Ref	Ref
	Midline	2385	402	16.9	1.43	(1.037, 1.967)	0.029 *
	Endline	3037	571	18.8	1.43	(0.929, 2.188)	0.105

Model controlled for participant age, education, and wealth. OR = Odds Ration, CI = Confidence Interval, LRT = Likelihood Ratio Test, Ref = Reference. * = Statistically significant.

## Data Availability

Restrictions apply to the availability of these data. Data was obtained from Corus International/IMA World Health and are available from the authors with the permission of Corus International/IMA World Health.

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
