# Peer review of "The Impact of a Large-Scale Social and Behavior Change Communication Intervention in the Lake Zone Region of Tanzania on Knowledge, Attitudes, and Practices Related to Stunting Prevention"

_ijerph, 2023, doi:10.3390/ijerph20021214_

Round 1
Reviewer 1 Report (Previous Reviewer 2)
Thank you the authors for addressing all my concerns. I am satisfied with the changes made, and happy to suggest this manuscript to be accepted after correcting for some typing and grammar mistakes.
Author Response
Thank you for the helpful feedback. The manuscript has now been reviewed and edited for grammar and spelling. Track changes has been used.
This manuscript is a resubmission of an earlier submission. The following is a list of the peer review reports and author responses from that submission.
Round 1
Reviewer 1 Report
Introduction
The introduction is very well written and provides a lot of details. Seeing as this study reports on a database that has other published papers, it would be great to note how these parameters of SBCC impacted stunting. It has a lot of data on SBCC but not how it impacts stunting, which is also a part of your narrative, make that clear in the introduction instead of just mentioning the previous studies from references 9-18.
Methodology
I suggest you include how the socio-economic status was calculated for reproducibility’s sake under wealth measures. This means the actual equation of the grading to get the scores.
Something that is not clear in your methodology is whether or not the same participants were interviewed at bassline, midline or endline. It seems to suggest they were different participants for we see higher number at follow up than at baseline for men. Spell this out clearly.
Results
In table 3, Agree child should only be given breastmilk for first 6 months and Report 6 months when asked when child should be given other foods/liquids are similar outcomes with possible similar answers as they are essentially asking the same thing, explain how one is statistically significantly different. At least investigate why this would be the case and report on it.
Discussion
Rephrase sentence one of paragraph 3 of the discussion, it is not very clear, “Ideally, there would be consistent improvement in the practice of all targeted behaviour’s over time. Variations in improvement of different indicators at different time periods in this study, however, varying levels of success among indicators is common when evaluating large-scale SBCC interventions”.
General comments
This is a very well written paper and largely clear on concepts as well as scientific knowledge.
After reading the other papers that analysed the same database, I believe that this would be a stronger discussion if you reported your findings in relation to what previous authors have reported for it is the same narrative. Consider incorporating these instead of only looking at the SBCC as stand alone, otherwise your topic is misleading because the reader would assume that you will have some information on the stunting outcome in relation to the SBCC’s.
The number of participants that already exposure to KAP was high in these populations, this could have impacted what was observed, explore adding more data on that in relation to other SBCC studies.
Do include recommendations on what other studies can focus on that has a larger impact.
Reviewer 2 Report
This manuscript assesses and demonstrates the positive impacts of a large-scale social and behaviour change communication (SBCC) intervention on female caregivers’ and male head of households’ KAP related to stunting prevention. This topic is of public interest, however there are few concerns that need to be addressed, see my comments as below:
Abstract:
- The extent of the changes in the KAP e.g., effect size of ORs should be reported in the abstract.
Methods:
- Does same respondents were assessed at three different time point? I assume they were different individuals. This needs to be clearly described.
- To compute the household wealth index, only two indices were included: i) access to services and ii) consumer durables. Why was housing quality not included? Is it valid to only use two of the indices and does it reflect the true household wealth index? This needs to be justified or at least explained.
Results
- Table 1: How about the characteristics of the male head of household?
- Table 2: At which time point that this intervention exposure is measured? Is it midline or endline? Please clarify.
- Table 5: Please recheck the results for “Perceive that half or more men in community help wives with chores during pregnancy”, it seems like there is significant change from baseline to endline (18.8%), given that percentage is higher than 16.9% in midline.
- For “How soon after birth should a child be put to the breast”, what is the cut off (months) used? Please clarify.
- All tables: Please define LRT. All the tables should be able to standalone without the main text, please provide the information on statistical analysis used and which are the reference groups used in the logistic regression e.g., baseline.
- It is still worth to assess the difference in the changes of KAP between those who exposed or not exposed (~30% of the study population) to the SBCC intervention, as well as through different exposure (e.g., IPC only, media only or both). This might help us to better contextualize the impacts of the SBCC approaches on KAP.
Discussion
- This sentence is quite misleading “For example, of the eight knowledge and attitudes indicators, four (50%) significantly improved at midline and four (50%) significantly improved by endline”, it sounds like all the eight indicators showed improvement either at midline or endline, but it is not in this case. Please rephrase the sentence, and this applies to the other KAP as well.
Conclusion
- The conclusion should summarize and highlight which KAP indicators that the SBCC approaches impacted the most, and which KAP indicators are more resistant to changes or less likely to be improved to inform future intervention implementation.
